# Predicting reliability through structured expert elicitation with the repliCATS (Collaborative Assessments for Trustworthy Science) process

Hannah Fraser[1‡], Martin Bush[1‡*], Bonnie C. Wintle[1,2], Fallon Mody[1], Eden T. Smith[1], Anca M. Hanea[1,3], Elliot Gould[1,2,3], Victoria Hemming[1,4], Daniel G. Hamilton[1], Libby Rumpff[1,2], David P. Wilkinson[1,2], Ross Pearson[1], Felix Singleton Thorn[1], Raquel Ashton[1], Aaron Willcox[1], Charles T. Gray[1,5], Andrew Head[1], Melissa Ross[1], Rebecca Groenewegen[1,2], Alexandru Marcoci[6], Ans Vercammen[7,8], Timothy H. Parker[9], Rink Hoekstra[10], Shinichi Nakagawa[11], David R. Mandel[12], Don van Ravenzwaaij[10], Marissa McBride[7], Richard O. Sinnott[13], Peter Vesk[1,2], Mark Burgman[7], Fiona Fidler[1]

1 MetaMelb Lab, University of Melbourne, Melbourne, Victoria, Australia, 2 Quantitative & Applied Ecology Group, University of Melbourne, Melbourne, Victoria, Australia, 3 Centre of Excellence for Biosecurity Risk Analysis, University of Melbourne, Melbourne, Victoria, Australia, 4 Martin Conservation Decisions Lab, Department of Forest and Conservation Sciences, University of British Columbia, Vancouver, Canada, 5 School of Natural and Environmental Sciences, Newcastle University, Newcastle upon Tyne, United Kingdom, 6 Centre for Argument Technology, School of Science and Engineering, University of Dundee, Dundee, United Kingdom, 7 Centre for Environmental Policy, Imperial College London, London, United Kingdom, 8 School of Communication and Arts, Faculty of Humanities and Social Sciences, The University of Queensland, Brisbane, Australia, 9 Department of Biology, Whitman College, Walla Walla, Washington, United States of America, 10 Faculty of Behavioural and Social Sciences, University of Groningen, Groningen, The Netherlands, 11 School of Biological, Earth and Environmental Sciences, University of New South Wales, Sydney, New South Wales, Australia, 12 Department of Psychology, York University, Toronto, Ontario, Canada, 13 Melbourne eResearch Group, University of Melbourne, Melbourne, Victoria, Australia

‡ HF and MB are joint first authors on this work.
* martin.bush@unimelb.edu.au

**Data Availability Statement:** The anonymized data underlying the validation experiment for the method described in this paper are available from a

## Abstract

As replications of individual studies are resource intensive, techniques for predicting the replicability are required. We introduce the repliCATS (Collaborative Assessments for Trustworthy Science) process, a new method for eliciting expert predictions about the replicability of research. This process is a structured expert elicitation approach based on a modified Delphi technique applied to the evaluation of research claims in social and behavioural sciences. The utility of processes to predict replicability is their capacity to test scientific claims without the costs of full replication. Experimental data supports the validity of this process, with a validation study producing a classification accuracy of 84% and an Area Under the Curve of 0.94, meeting or exceeding the accuracy of other techniques used to predict replicability. The repliCATS process provides other benefits. It is highly scalable, able to be deployed for both rapid assessment of small numbers of claims, and assessment of high volumes of claims over an extended period through an online elicitation platform, having been used to assess 3000 research claims over an 18 month period. It is available to be implemented in a range of ways and we describe one such implementation. An important advantage of the repliCATS process is that it collects qualitative data that has the potential

publicly accessible OSF repository (at https://osf.io/
hkmv3/). This link is provided in the manuscript.

**Funding:** This project is sponsored by the Defense
Advanced Research Projects Agency (DARPA)
under cooperative agreement No.
HR001118S0047. The content of the information
does not necessarily reflect the position or the
policy of the Government, and no official
endorsement should be inferred. The funders had
no role in study design, data collection and
analysis, decision to publish, or preparation of the
manuscript.

**Competing interests:** The authors have declared
that no competing interests exist.

to provide insight in understanding the limits of generalizability of scientific claims. The pri-
mary limitation of the repliCATS process is its reliance on human-derived predictions with
consequent costs in terms of participant fatigue although careful design can minimise these
costs. The repliCATS process has potential applications in alternative peer review and in
the allocation of effort for replication studies.

## 1. Introduction

Scientific claims should be held to a strong standard. The strongest claims will fulfil multiple
criteria. They will be *reliable* and reproducible, in that multiple observers examining the same
data should agree on the facts and the results of analyses, they will be *replicable*, in that repeti-
tions of the methods and procedures should produce the same facts/results, and they will be
*generalizable*, in that claims should extend beyond a single dataset or process. Evaluating the
strength of individual scientific claims on the basis of any one of these criteria, however, is not
straightforward, let alone all of them.

The focus of the work underlying this paper is on assessing the replicability of published
research claims. In assessing the strengths of scientific claims from different experimental and/
or disciplinary contexts the criteria listed above will be of varying importance, however, repli-
cation studies are clearly a key technique for assessing certain kinds of research and the resul-
tant literature. Such studies thereby contribute to the progressive development of robust
knowledge with respect to both inferences from empirical data and deductions made from
empirical evidence; policy-making and public trust both draw on such developments. Interest-
ingly, several large-scale replication projects in the social sciences and other disciplines have
shown that many published claims do not replicate [1–4]. These failures to replicate may indi-
cate false positives in the original study, or they may have other explanations, such as differ-
ences in statistical power between the original and the replication, or unknown moderators in
either.

This paper outlines an expert elicitation protocol designed to evaluate the replicability of a
large volume of claims across the social and behavioural sciences. We describe the utility, accu-
racy and scalability of this method, how it has been validated, and discuss how this technique
has the potential to provide insight into aspects of the credibility of scientific claims beyond
replicability, such as the generalizability of claims.

The elicitation procedure in this paper a modified Delphi process called the IDEA protocol
[5, 6], and involves experts working in small groups to provide quantitative predictions, and
associated (qualitative) reasoning of the probability of successful replication. The qualitative
data provides information about participants' judgements, including about the credibility sig-
nals of claims beyond replicability.

The IDEA protocol has been applied to assessing the replicability of research claims in the
social and behavioural sciences by the repliCATS (Collaborative Assessments for Trustworthy
Science) project as a component of the Systematizing Confidence in Open Research and Evi-
dence (SCORE) program funded by the US Defence Advanced Research Projects Agency. The
overall goal of the SCORE program is to create automated tools for forecasting the replicability
of research claims made within the social and behavioural science literature [7]. In Phase 1 of
the SCORE program, we elicited expert assessments of replicability for 3000 research claims.
These assessments formed a benchmark for the comparison of the performance of automated
tools developed by other teams within the SCORE program. In Phase 2 of the SCORE program

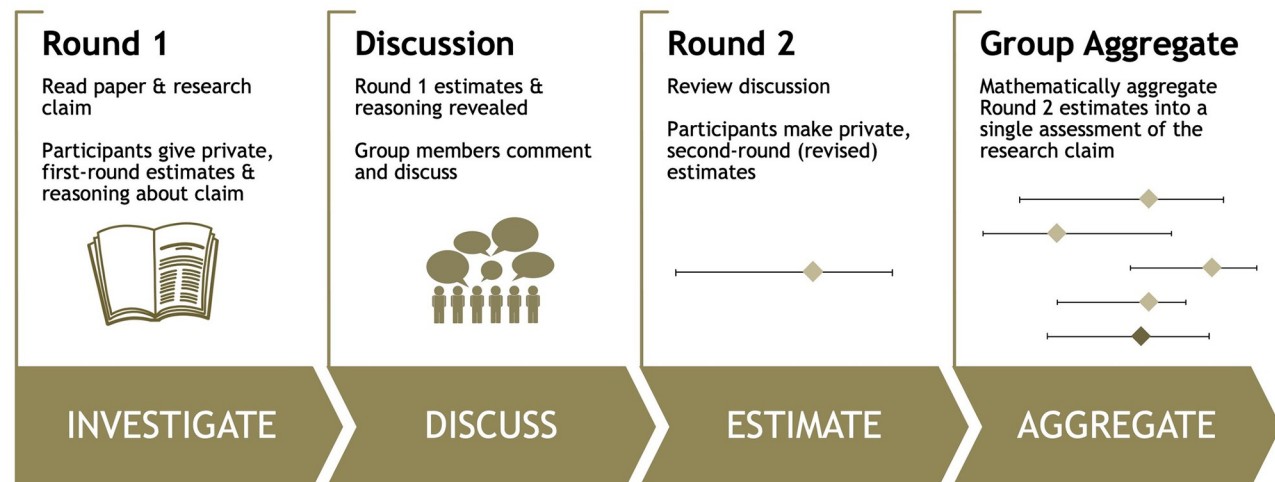

**Fig 1. Overview of the IDEA protocol, as adopted in the repliCATS project.**

we expanded the elicit assessments of replicability for another 900 specific research claims while expanding the elicitation to include additional credibility signals of 200 papers.

The repliCATS project introduces a new approach to predicting the replicability of research claims. It is neither a prediction market, nor a simple one-off survey. The repliCATS approach, involves four steps, represented in the acronym IDEA: 'Investigate', 'Discuss', 'Estimate' and 'Aggregate' (Fig 1). Each individual is provided a scientific claim and the original research paper to read, and provide an estimate of whether or not the claim will replicate (*Investigate*). They then see the group's judgements and reasoning, and can interrogate these (*Discuss*). Following this, each individual provides a second private assessment (*Estimate*). A mathematical aggregation of the individual estimates is taken as the final assessment (*Aggregate*).

## 2. Utility, accuracy and validity, scalability, and insight

### 2.1 Utility

Whilst replication studies have significant epistemic value, they are expensive, labour intensive, and face logistical barriers, such as holding little career incentives. In other cases, they are simply not feasible at all. For example, national censuses are not feasibly recreated and historical circumstances are impossible to study again. Even where the contextual factors are more favourable, the expense of full replication studies requires analysis of the potential benefits against costs [8]. Therefore, analytical techniques for a prognostic assessment of the reliability, replicability and generalizability of research claims without attempting full replications have substantial utility because they can provide similar benefits at much lower cost. They can also, in some cases, inform decisions about where to direct scarce resources for such full replications [8, 9]. The method described in this paper is one such process, and thereby provides this kind of cost-saving utility.

### 2.2 Accuracy and validity

To measure the accuracy of the repliCATS process, we compare it with existing techniques for predicting the outcomes of replication studies. The two main techniques that have been used for human-derived predictions are surveys and prediction markets [2, 10, 11]. In the former, experts give independent judgements which are aggregated into quantitative predictions of

**Table 1. Summary of different metrics commonly used to evaluate predictions.**

| Metric | Definition | Advantages | Disadvantages |
|---|---|---|---|
| Classification Accuracy | Percentage of assessments (probability estimates) >50% that successfully replicated, and <50% that did not successfully replicate | Easy to calculate. Intuitively understood. Comparable across datasets | Loses information |
| Area Under the Curve (AUC) | Obtained by plotting the true positive rate against the false positive rate. The best possible predictor with no false negatives and no false positives produces a point at (0,1), and a random assessment gives a point along the diagonal | Does not rely on a single cutoff, considers all possible thresholds | Loses information |
| Brier Score | The squared difference between an estimated probability (a participant's best estimate) and the actual outcome | Uses all the information. Easily interpretable as a mean square error | Not easily comparable across different datasets. Only useful for long term accuracy |
| Informativeness | The average width of the intervals between the upper and lower bounds elicited alongside best estimates | Does not require performance information | Not properly operationalised for in a (classical) probabilistic framework |

replicability. In the latter, participants trade contracts that give a small payoff if and only if the study is replicated and prices are used to derive the likelihood of replicability. Both of these have typically been implemented alongside large-scale replication projects and used replication outcomes as the ground truth to test prediction accuracy. There are other techniques for predicting replicability, such as machine learning methods [12, 13], but such fully automated techniques are not considered further in this paper.

There are multiple ways of measuring the success of predictions (Table 1). The most straightforward is 'classification accuracy'. This treats predictions >50% as predictions of replication success and <50% as predictions of replication failure. Classification accuracy is the percentage of predictions that were correct (i.e. on the right side of 50%), excluding predictions of 50%.

Existing techniques for predicting replicability perform well. The classification accuracy from previous prediction studies has ranged between 61% and 86%. The lower limit was reported by Camerer et al. [1] for both surveys and prediction markets on the replicability of 18 laboratory experiments in economics. The higher limit was from Camerer et al. [2] for surveys and prediction markets on predicting the replicability of 21 social science experimental studies published in Nature and Science between 2010 and 2015. In both studies, surveys and prediction markets performed well. Other studies have reported small but noteworthy differences. For the surveys and markets running alongside the Many Labs 2 project [3], prediction markets had a 75% classification accuracy while pre-market surveys reached 67%. There is also evidence that for some kinds of social science research claims non-experts are able to make fairly accurate predictions for the replicability of research claims [14].

An essential goal of the repliCATS project is to improve upon this already good accuracy of current techniques for predicting replicability. Fine-tuning expert accuracy in forecasting replicability is a non-trivial challenge. The primary ways in which the IDEA protocol is expected to meet the exacting demands of this task are by harnessing the wisdom of the crowd and diverse ideas, sharing information between participants to resolve misunderstandings and promote further counterfactual thinking, and by taking an aggregate of the judgement. These help to improve accuracy, and reduce overconfidence by reducing biases such as anchoring, groupthink, and confirmation bias. The IDEA protocol aims to improve accuracy through controlling groupthink by aggregating group assessments *mathematically* and not *behaviourally* [15–18]. That is, group members are not forced to agree on a single final judgement that reflects the whole group. Mathematical aggregation can be more complex than taking the arithmetic mean. Several aggregation techniques have been pre-registered by repliCATS (https://osf.io/

5rj76/). Our many mathematical aggregation models are in detailed in Hanea et al. [19], these fall into three main groupings: 1) linear combinations of best estimates, transformed best estimates [20] and distributions [21]; 2) Bayesian approaches, one of which incorporates characteristics of a claim directly from the paper, such as sample size and effect size [22]; and 3) linear combinations of best estimates, mainly weighted by potential proxies for good forecasting performance, such as demonstrated breadth of reasoning, engagement in the task, openness to changing opinion, informativeness of judgements, and prior knowledge (inspired by Mellers et al. [23, 24]). Groupthink is also controlled by recruiting groups that are, ideally, as diverse as possible [25]. Work in structured elicitation has described how the sharing of information can improve the accuracy of group judgements [26]. The IDEA protocol implemented by the repliCATS process implements this feature, in contrast with survey-based methods of prediction, which generally do not allow for the sharing of information between participants. The IDEA protocol also reduces overconfidence in individual judgements through the use of three-point elicitations, that is asking participants to provide lower and upper bounds for their assessment, as well as their best estimate [27, 28]. This technique [29, 30] is thought to encourage information sampling [29] and prompt participants to consider counter-arguments [31]. However, see also [32] for counter-evidence regarding this sequencing.

To test the quantitative predictions of the replicability of research claims produced by the repliCATS process, we conducted a validation experiment [33] alongside the SIPS conference in 2019. This experiment was conducted independently of the SCORE program. A pre-registration for this experiment can be seen at https://osf.io/e8dh7/. Five groups of five participants separately assessed the replicability of 25 claims drawn from previous replication projects, including projects that had run prediction studies using alternative methods for predicting replicability. A summary of results from this experiment is provided in Fig 2, showing all group assessments for each of the 25 claims. The specific elicitation used appears in the S1 Table. The full anonymized dataset from this validation experiment is available at https://osf.io/hkmv3/.

When all judgements from the validation experiment were aggregated, participants using the repliCATS process achieved a classification accuracy of 84% (and an *Area Under the Curve* measure of 0.94). Compared with published accuracies for other processes, [1–3] this indicates that the repliCATS technique performed comparably or better than existing methods for predicting replicability. We rely on previously published accuracies for comparison with existing methods as it was not possible to directly test the performance of these five groups using alternative prediction techniques without duplicating the experiment and thus the burden on participants.

We note that the results of this validation study should be interpreted with caution. The majority of articles in this study were from psychology papers and most participants at the SIPS conference were psychology researchers. Thus we considered that they may have known more about the signals or replicability for these papers than a participant may have for a typical claim, for which they may have less disciplinary insight. It was also expected that in some instances participants were familiar with the results of replications for the claims they evaluated, as these results were publicly available prior to SIPS 2019. We infer that the study has validated the use of the repliCATS process for application but a more precise measurement of its accuracy will depend on further work.

## 2.3 Scalability

In addition to accuracy, repliCATS aims to provide a scalable process. The IDEA protocol is typically implemented through group sizes between four and seven, with perhaps one or two

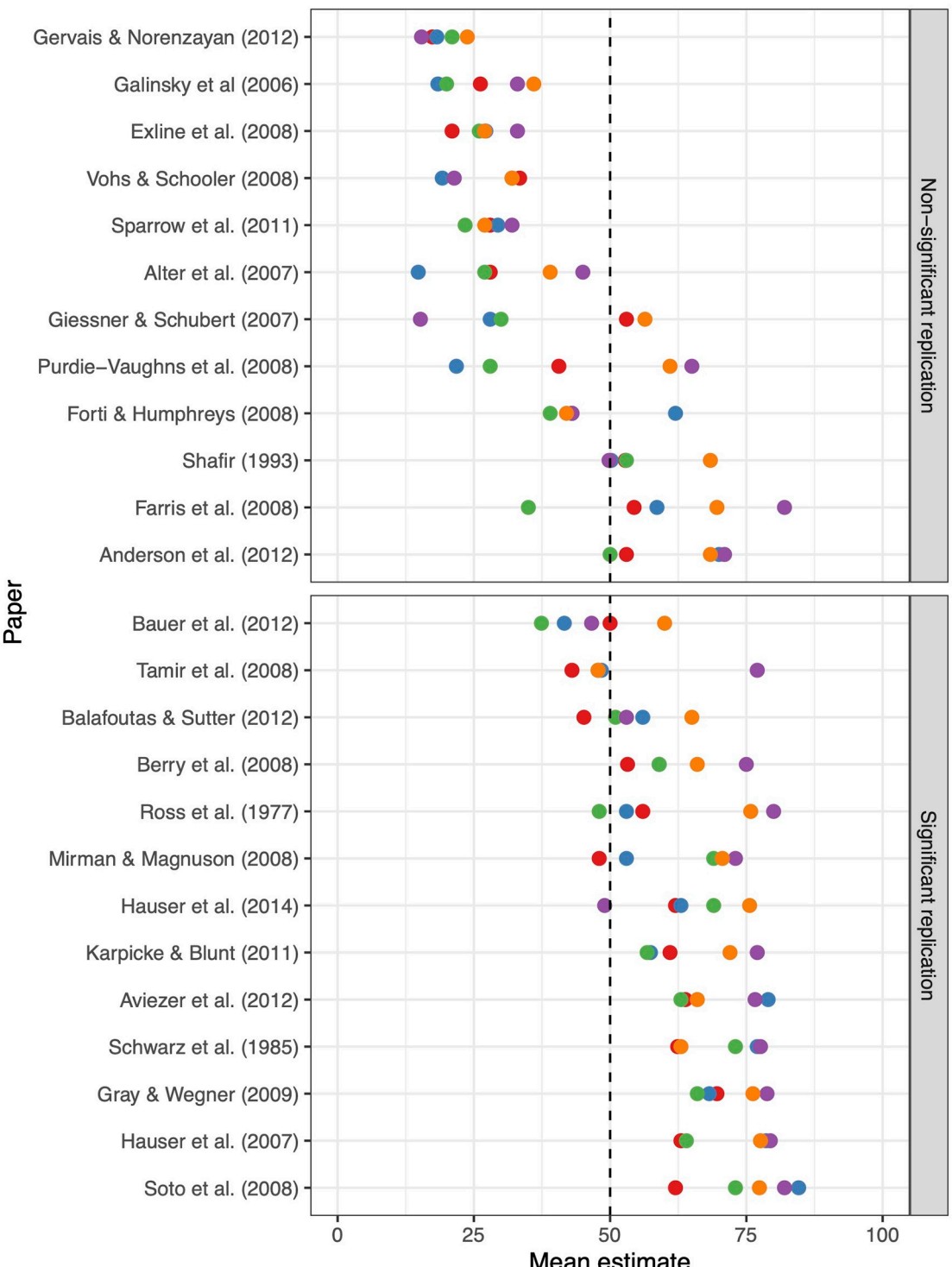

**Fig 2. Summary of results from validation experiment.** Each dot represents a particular group's aggregated assessment of replicability for each claim, with five groups assessing all 25 claims drawn from previous replication projects. Claims for which the replication study did not produce a statistically significant result are shown in the top pane, and claims for which the replication study did produce a statistically significant result are shown in the bottom pane.

groups of experts per problem, each working under the guidance of a facilitator. This process allows for a rapid evaluation of claims, in contrast with prediction markets that rely on many participants engaged in multiple trades. At the same time, an online platform implementation of the IDEA protocol allowed for many experts to address many problems. The technical details of this specific implementation of the repliCATS process are described by Pearson et al. [34]. However, the process described in this paper could be implemented through other realizations.

The primary application of the repliCATS process to date has been through the SCORE program. The scalability of this process is demonstrated by the volume of claims assessed as part of this program—3000 claim assessments in 18 months in Phase 1 of the SCORE project, which ran from February 2019 to 30 November 2020 and a further 900 claim assessments and 200 paper-level assessments in Phase 2, running from December 2020 to May 2022. Alongside this, the online platform developed for this program has been used in a number of experiments and classroom teaching contexts as briefly described below. In Phase 1 of the SCORE project, on which we concentrate, the repliCATS project was provided with 3000 social and behavioural science claims for evaluation by another team involved in the SCORE program. This independent team was also responsible for conducting replication studies for a subset of the 3900 claims. The claims were drawn from 62 peer-reviewed journals across eight social and behavioural science disciplines: business research, criminology, economics, education, political science, psychology, and sociology. Full details of these journals are provided in the S2 Table. The repliCATS project gathered expert predictions of replicability through a combination of face-to-face and asynchronous workshops, as well as fully remote elicitations. One third of the claims, as well as the data for the validation study presented below, were elicited in three large workshops held alongside relevant conferences. Two of the workshops were held alongside the Society for Improvement of Psychological Science (SIPS) conference in July 2019 (Netherlands) and May 2020 (virtual workshop); and the third was alongside the inaugural Association for Interdisciplinary Metaresearch & Open Science (AIMOS) conference in November 2019 (Melbourne). In workshops, participants were divided into small groups with a facilitator and assessed claims synchronously. Different workshop groups took differing amounts of time for each claim but on average assessments were completed in under 30 minutes.

## 2.4 Insight

The repliCATS process is capable of generating valuable qualitative data to provide insight on issues beyond the direct replicability of specific evidentiary claims. Such problems include: identifying the precise areas of concern for the replicability of a given claim; understanding the limits of generalizability and hence potential applicability of a claim for a research end-user, and assessing the quality of the operationalization of a given research study design. For example, a research claim may be highly replicable and yet offer a poorly operationalized test of the target hypothesis [35]. In such a case, knowing only the predicted replicability of the claim says little about the status of the overall hypothesis. Similarly, even if a claim is well-operationalized, understanding how applicable a claim is for research end-users means understanding its limits of generalizability. This aspect of the repliCATS project, as an implementation of the IDEA protocol, is arguably the most critical point of difference from previous approaches to predicting replicability. Although both surveys and prediction markets can be adapted to collect this kind of data, the advantage of the repliCATS process described here is that such data is produced directly through the elicitation itself, resulting in more straightforward and richer data generation, collection and subsequent analysis.

This elicited data describes participants' reasoning about the claims being evaluated, typically including justifications for the assessment of the target question about replicability, as well as judgements about the papers' importance, clarity and logical structure. While such reasoning data has been a part of previous applications of the IDEA protocol, it has not been the focus of previous studies. The repliCATS project thus extends the IDEA protocol through a structured qualitative analysis of a substantial corpus. This data has the potential to address a number of questions that are of interest for replication studies, for research end-users, and more broadly in metaresearch and the philosophy of the sciences. Some of these target problems, like understanding the quality of operationalization of a given research design, or the limits of generalizability of a particular claim, were described in Section 2.2 above. Other issues include identification of particular strengths or weaknesses in a given research claim, with implications for how a claim might be further investigated. Participants' justifications might indicate weaknesses with a particular kind of measurement technique, suggesting that further work should be focused on that aspect of experimental design. Alternatively, they might indicate a very high level of prior plausibility for the claim, in which case further work on the theoretical background of the claim might be more fruitful. There are many ways to address such qualitative data. Within the repliCATS project, qualitative analysis of data is briefly as follows. A subset of the data (776 of the set of 3000 claims in Phase 1 of the SCORE program, which comprised 13901 unique justifications from a total dataset of 46408 justifications) was coded by a team of five analysts against a shared codebook, with each justification being coded by at least two coders. Several principles were applied in developing the codebook for analysis. Inclusion and exclusion criteria were developed for codes most relevant to the research questions. In particular, this included both direct markers of replicability (e.g. statistical details) and proxy markers of replicability (e.g. clarity of writing). The total size of the codebook was limited to ensure its usability. The codebook was developed iteratively, with discussion among coders between each round of development to provide a form of contextualised content analysis. In addition, the inter-coder-reliability (ICR) for each code was assessed at various points in this process. The ICR results were used for reviewing the convergence of textual interpretation between analysts, and after coding, mixed-method aggregation techniques only utilised those codes that had met a pre-registered ICR.

## 3. Current implementations

The repliCATS process described here could be implemented in several ways, including a simple pen-and-paper version. We have developed a cloud-based, multi-user software platform ('the repliCATS platform') that supports both synchronous face-to-face workshops and asynchronous remote group elicitations [34]. A full description of the elicitation used on this platform for the validation experiment appears in the S1 Table. Fig 3 provides a snapshot of the technical operationalization of this elicitation as supported in the repliCATS platform. In particular, Fig 3 shows an example of aggregated group judgements and reasoning from round 1, as shown to participants in round 2 prior to submitting their final judgements and reasoning.

This specific repliCATS platform incorporated several features beyond the IDEA protocol that were designed to improve participants' confidence in completing their assessments, as well as support or enhance the elicitation process. These features include ready access to general decision support materials and a glossary; tooltips for each question to provide additional guidance on answering each question; additional interactive discussion elements such as the ability to upvote, downvote and have threaded comments; and gamification in the form of participant badges. The badges were designed to reward participation as well as to reward behaviours considered to be beneficial for improving participants' and group judgements. For

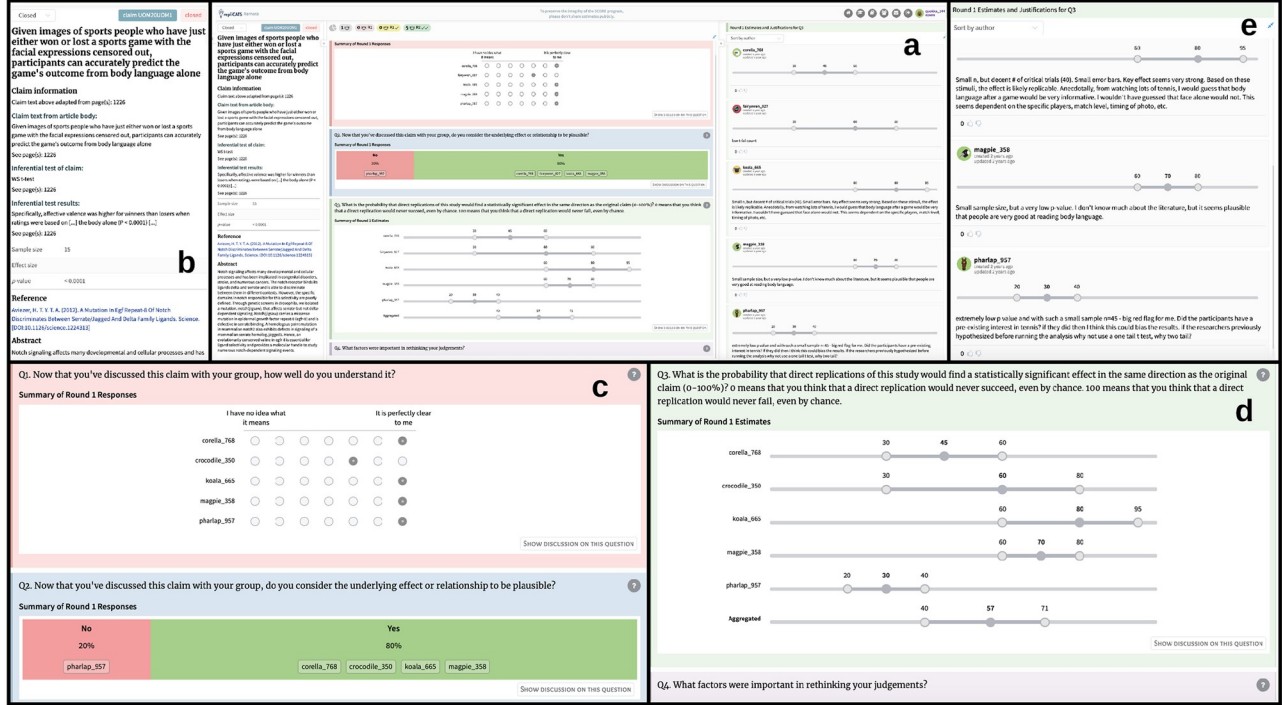

**Fig 3. The repliCATS platform.** Anticlockwise from centre top (a) complete layout, plus expanded details of: (b) claim information; (c) and (d) responses to elicitation and aggregated feedback; and (e) participants' reasoning comments.

example, participants were awarded badges for seeking out decision support materials, interacting with group members' reasoning, and consistently submitting their reasoning while evaluating claims.

This implementation was used in Phase 1 of the SCORE program by 759 registered participants. Of these, 550 users assessed one or more of the 3000 claims. All participants explicitly provided consent through the online platform and participant activities were approved by the Human Research Ethics Committee of The University of Melbourne (ID 1853445). However, some participants used multiple identities and not all student participants could be individually identified due to the additional ethics clearance obtained from the University of North Carolina for the teaching use of the repliCATS platform described in section 5.1.

## 4. Limitations of the repliCATS process

The advantages of the IDEA process also come with challenges due to issues of elicitation, including recruitment of participants, that are likely to be generally applicable to most implementations of the repliCATS process. Other limitations faced by the repliCATS project were based in the specific program context in which this process was developed and implemented. These will be relevant for some but not all implementations of the repliCATS process and will not be discussed further beyond noting that principally amongst these was the inability, within the recruitment burden, for claims to be assessed by multiple groups.

### 4.1 Challenges with elicitation

A problem for any research involving human subjects is elicitation burden—both the quantity and quality of information provided by experts are thought to decline the longer the elicitation

process takes, similar to participant fatigue in surveys and experiments [36]. There is thus a trade-off between the extent of information desired for research purposes and the number and richness of questions asked. As noted, the average time taken for a group to assess a claim in repliCATS workshops was just under 30 minutes. Such a bounded time of participation was also necessary because the majority of research participants were volunteers who were unpaid, or only minimally compensated for their time. Although aspects of the elicitation are believed to be intrinsically rewarding—by design—this is another reason for minimizing the burden on participants.

In order to address the elicitation burden, the repliCATS platform was designed to be user-friendly, and online workshops were flexible in terms of participant schedules. However, the trade-off for such elicitations is that it reduced the capacity for participants to engage in discussion, which is a key aspect of the IDEA protocol. Participants in widely-spaced time zones could not have synchronous discussions, and could only exchange comments over a period of days. Other online workshops using the repliCATS process have grouped participants, as best as possible into compatible time zones, although other logistical constraints do not always make this possible. (By contrast, face-to-face workshops allow for full discussions, at the cost of committing participants to a set schedule and physical co-location.) Given that the information-gathering phase of the process constituted around one-third of the time spent by participants (approximately 30 minutes per claim assessed), it is difficult to imagine a method that will substantially reduce this burden. Prediction markets have the potential for shorter engagement but only if participants do not return to engage in repeated trades. In this case, as with all methods, there is a trade-off between effort by participants and performance. Ultimately, the problem of participant fatigue is inherent to using human-derived judgements which is best addressed by user-centred design appropriate to the particular method.

A related difficulty is that research participants might have multiple interpretations of key terms for the elicitation. Indeed, there can be considerable conceptual slippage around many of these terms, such as conflicting taxonomies attempting to define replication practices. While the discussion phase of the IDEA protocol can be used to work on resolving such ambiguities, it can be difficult or impossible to completely eliminate such differences in interpretation. This difficulty was addressed by encouraging participants to describe their interpretations through textual responses, allowing both team members and qualitative analysts to understand these ambiguities better. The repliCATS platform also provides the definitions for key terms such as 'successful replication' where specific definitions are used within the SCORE program. The provision of such information also involves trade-offs between definitional specificity and richness of response. This problem of ambiguity is inherent to human assessment of claims and so is relevant to alternative techniques, although the impacts will be different for each technique.

Another challenge is that participants may hold opinions about aspects of a study that could bias their judgements about replicability. One such issue relates to the perceived importance of a study, whilst another relates to stylistic features of how the paper is written. While both of these may be proxies for the replicability of a study in some contexts, they need not be. Even more directly relevant markers, like the quality of the experimental design, need not suggest non-replicability. For example, a poorly operationalized study where the dependent variable is auto-correlated with the independent variable will, in fact, be highly replicable if you repeat the study exactly. For these reasons, the elicitation was designed to separate out judgements about replicability while allowing participants to explicitly express their opinions about matters like the clarity, plausibility and importance of the study. This is done through two questions prior to the question about replicability, and one question afterwards. Like the problem of ambiguity, this challenge affects alternative techniques for predicting replicability,

although, also like the previous case, the discussion phase of the IDEA protocol has the advantage of allowing the problem to be directly addressed.

## 4.2 Challenges with recruitment

Ideally, for most forecasting problems, IDEA groups are deliberately constructed to be diverse. People differ in the way they perceive and analyze problems, as well as the knowledge they bring to bear on them. Access to a greater variety of information and analyses may improve individual judgements. The accuracy of participants' answers to quantitative and probabilistic judgements has been shown to improve between peoples' independent initial judgements and the final judgements they submit through the IDEA protocol [5, 6]. Offset against this is that domain-specific knowledge is clearly relevant to understanding the details of research claims and thus being able to assess their replicability. The balance between diversity and domain knowledge in good assessments of replicability is poorly understood. This problem warrants further research. The repliCATS recruitment strategy meant that we were not able fully to control for or consistently recruit diversity in groups. Where participants are allowed to self-select claims to assess, such diversity is even harder to achieve, even if the overall participant pool contains a large amount of diversity. Maximising diversity will be a challenge for any implementation of the repliCATS process, subject to the specific constraints of the implementation context.

## 5. Potential applications

The repliCATS process is intended to be used by researchers well beyond its application to the SCORE program. We describe potential application of this method that we intend and that may interest others.

### 5.1 Capacity building in peer review

There are few peer review training opportunities available to researchers [37] and even fewer that have clearly demonstrated evidence of success [38, 39]. As a consequence, many early career researchers report that they frequently learn how to review in passive and fragmentary ways, such as performing joint reviews with their advisors and senior colleagues, e.g. via participation in journal clubs or from studying reviews of their own submissions [40].

Participants in the repliCATS project noted the benefits of the feedback and calibration in the repliCATS process: "I got a lot of exposure to a variety of research designs and approaches (including some fun and interesting theories!) and was afforded [the] opportunity to practice evaluating evidence. In practicing evidence evaluation, I feel like I sharpened my own critical evaluative skills and learned from the evaluations of others." This type of feedback was particularly common from early career researchers, although more experienced researchers also described the value of the process as professional development in peer review.

This feedback suggests the potential to deploy evaluations of research papers through the repliCATS process as explicit training in both research design and peer review. A pilot application of such student training has already been undertaken at the University of North Carolina, Chapel Hill where the repliCATS process was deployed as an extra credit undergraduate student activity.

### 5.2 Alternative peer review models

Surprisingly little is known about how reviewers conduct peer review. More than a quarter of a century after the remark was made, it is still the case that "we know surprisingly little about the

cognitive aspects of what a reviewer does when he or she assesses a study" [41]. Nor are journal instructions to reviewers especially informative here with criteria for acceptance usually expressed in general terms, such as "how do you rate the quality of the work?".

Initiatives such as Transparency and Openness Promotion (TOP) guidelines offer guidance for authors and reviewers at signatory journals [42] (see also https://www.cos.io/initiatives/top-guidelines). These are important for ensuring completeness of scientific reporting. The repliCATS process is not an attempt at competing guidelines. Its goal is not to provide checklists for reviewers (or authors or editors). Rather, we conceptualize peer review as a process that takes advantage of collective intelligence through an expert deliberation and decision-making process, and thus one to which a structured elicitation and decision protocol is applicable [43].

Despite limited implementation among popular journals in the social sciences [44], there are existing models of interactive peer review that encourage increased dialogue between reviewers, such as the one used in the Frontiers journals. However, these typically rely on behavioural consensus, with its attendant disadvantages. In contrast, the repliCATS process has the strong advantage of a predefined end-point, avoiding 'consensus by fatigue'. It is transparent by design, and the underlying IDEA protocol is directly informed by developments in the expert elicitation, deliberation and decision making literature.

## 5.3 Commissioned reviews

One particular example of alternative forms of review includes commissioned reviews, such as for papers or research proposals prior to submission. This potential application was suggested by a number of repliCATS participants. Variations on this theme include reviews of a suite of existing published research intended as the basis of a research project or as the evidence base for specific policy, action or management decisions. The former is likely to appeal to early career researchers, for example, at the start of their PhD candidature, and the latter to end users and consumers of research. The scalability of the repliCATS process makes it suitable for this use.

## 5.4 Allocating replication effort

Replication of studies is not always possible nor is it always desirable. Some studies cannot be replicated, as described above, due to, for example, their historical nature. Such studies may still have the potential to inform decisions and assessment of their reliability may still be valuable. Nor is replication always ideally suited for assessing the reliability of a claim, even when it is a viable approach. In any case, replications are typically resource-intensive.

There are several approaches to determining how best to allocate scarce resources to replication studies. One suggested approach is to apply the results of prediction markets to this question [10]. Other approaches propose selection of studies for replication based on a trade-off between the existing strength of evidence for a focal effect and the utility of replicating a given study. In Field et al. [45], a worked example is given on how to combine strength of evidence (quantified with a Bayesian reanalysis of published studies) with theoretical and methodological considerations. Pittelkow et al. [46] follow a similar procedure, applied to studies from clinical psychology. Isager et al. [9] outline a model for deciding on the utility of replicating a study, given known costs, by calculating the value of a claim and the uncertainty about that claim prior to replication. The key variables in this model—costs, value, and uncertainty—remain undefined, with the expectation that each can be specified outside the model (as relevant to a given knowledge domain). This approach to formalizing decisions can help clarify how to justify allocating resources toward specific replication practices.

While all techniques for predicting replicability can generate data about specific research claims that can be used as inputs to a formal model can provide further inputs. For example, the repliCATS approach is able to provide data on uncertainty, such as the extent of agreement or disagreement within groups about the replicability of a given claim, and the potential uncertainty of individual assessments as seen in interval widths. Additionally, qualitative data from the repliCATS process provides information on the theoretical or practical value of specific claims.

## 6. Conclusion

Assessment of the credibility of scientific papers in general, and predicting the replicability of published research claims in particular is an expert decision-making problem of strong importance to the scientific community. In such cases the use of a structured protocol has known advantages. The repliCATS process is a practical realization of the IDEA protocol for the assessment of published research papers. A validation experiment has shown that this process meets or exceeds the accuracy of current techniques for predicting replicability. The repliCATS process has also been demonstrated to be scalable, from undertaking rapid review of small numbers of claims to the capacity to assess a large volume of claims over an extended period. A particular advantage of the repliCATS process is the collection of rich qualitative data that can be used to address questions beyond those of direct replicability, such as the generalizability of given claims. Future work building on the repliCATS process to incorporate broader sets of credibility signals beyond replicability and in a range of applications has the potential to improve the scientific literature by supporting alternative models of peer review.

## Supporting information

**S1 Table. Details of elicitation questions used in the repliCATS process for the validation experiment.**
(TIF)

**S2 Table. List of source journals for research claims in the SCORE program.**
(TIF)

## Author Contributions

**Conceptualization:** Hannah Fraser, Martin Bush, Bonnie C. Wintle, Fallon Mody, Eden T. Smith, Anca M. Hanea, Elliot Gould, Victoria Hemming, Libby Rumpff, Ross Pearson, Felix Singleton Thorn, Alexandru Marcoci, Ans Vercammen, Timothy H. Parker, Rink Hoekstra, Shinichi Nakagawa, Peter Vesk, Mark Burgman, Fiona Fidler.

**Data curation:** Elliot Gould, David P. Wilkinson, Ross Pearson, Felix Singleton Thorn, Aaron Willcox, Charles T. Gray, Rebecca Groenewegen, Richard O. Sinnott.

**Formal analysis:** Hannah Fraser, Martin Bush, Bonnie C. Wintle, Fallon Mody, Eden T. Smith, Anca M. Hanea, Elliot Gould, David P. Wilkinson, Felix Singleton Thorn, Aaron Willcox, Ans Vercammen, Don van Ravenzwaaij, Marissa McBride, Fiona Fidler.

**Funding acquisition:** Fiona Fidler.

**Investigation:** Hannah Fraser, Martin Bush, Bonnie C. Wintle, Fallon Mody, Eden T. Smith, Anca M. Hanea, Elliot Gould, Victoria Hemming, Daniel G. Hamilton, Libby Rumpff, David P. Wilkinson, Ross Pearson, Felix Singleton Thorn, Raquel Ashton, Aaron Willcox, Charles T. Gray, Andrew Head, Melissa Ross, Alexandru Marcoci, Ans Vercammen,

Timothy H. Parker, Rink Hoekstra, Shinichi Nakagawa, Don van Ravenzwaaij, Marissa McBride, Peter Vesk, Mark Burgman, Fiona Fidler.

**Methodology:** Hannah Fraser, Martin Bush, Bonnie C. Wintle, Fallon Mody, Eden T. Smith, Anca M. Hanea, Elliot Gould, Victoria Hemming, Libby Rumpff, Ross Pearson, Felix Singleton Thorn, Alexandru Marcoci, Ans Vercammen, Timothy H. Parker, Rink Hoekstra, Shinichi Nakagawa, David R. Mandel, Peter Vesk, Mark Burgman, Fiona Fidler.

**Project administration:** Hannah Fraser, Melissa Ross, Fiona Fidler.

**Software:** Richard O. Sinnott.

**Supervision:** Fiona Fidler.

**Writing – original draft:** Hannah Fraser, Martin Bush, Bonnie C. Wintle, Fallon Mody, Eden T. Smith, Anca M. Hanea, Elliot Gould, Victoria Hemming, Daniel G. Hamilton, Libby Rumpff, Melissa Ross, Alexandru Marcoci, Ans Vercammen, Timothy H. Parker, Rink Hoekstra, Shinichi Nakagawa, David R. Mandel, Don van Ravenzwaaij, Richard O. Sinnott, Mark Burgman, Fiona Fidler.

**Writing – review & editing:** Martin Bush, Fiona Fidler.

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
