## [Decision Letter · Decision Letter 0]

4 Jun 2021

PONE-D-21-05317

Predicting reliability through structured expert elicitation with repliCATS (Collaborative Assessments for Trustworthy Science

PLOS ONE

Dear Dr. Bush,

Thank you for submitting your manuscript to PLOS ONE. After careful consideration, we have decided that your manuscript does not meet our criteria for publication and must therefore be rejected.

I am sorry that we cannot be more positive on this occasion, but hope that you appreciate the reasons for this decision.

Yours sincerely,

Sherief Ghozy, M.D., Ph.D. candidate

Academic Editor

PLOS ONE

Reviewers' comments:

Reviewer's Responses to Questions

**Comments to the Author**

1. Is the manuscript technically sound, and do the data support the conclusions?

Reviewer #1: No

2. Has the statistical analysis been performed appropriately and rigorously? 

Reviewer #1: N/A

3. Have the authors made all data underlying the findings in their manuscript fully available?

Reviewer #1: No

4. Is the manuscript presented in an intelligible fashion and written in standard English?

Reviewer #1: Yes

5. Review Comments to the Author

Reviewer #1: This paper “outlines an expert elicitation protocol (repliCATS) designed to accurately predict the replicability of a large volume of claims across the social and behavioural sciences.” Its main research goals are: accuracy, scalability and insight. Nowhere is the claim of accuracy supported by argument, data or even elaborated. As far as I can see the paper consists only of describing their platform and their aspirations. There is no scientific claim to review and nothing for a reviewer to do except to say that this manuscript doesn’t belong in a scientific journal. Indeed it reads more like an interim report to a funding agency. I will say that judging “classification accuracy” by treating “predictions >50% as predictions of replication success and <50% as predictions of replication failure” is extremely maladroit. Why go to the trouble of eliciting probabilistic predictions?

6. PLOS authors have the option to publish the peer review history of their article (what does this mean?). If published, this will include your full peer review and any attached files.

Reviewer #1: No

- - - - -

---

## [Author Response · Author response to Decision Letter 0]

14 Oct 2021

Responses to the reviewer were included in the PointResponses attachment. I copy them below. 

Reviewer’s comments 

“Nowhere is the claim of accuracy supported by argument, data or even elaborated.”

As a Methods paper, we are required to provide validation of our method. This was done in §4. We welcome comments that will improve this discussion. We disagree that validation was nowhere supported.

“As far as I can see the paper consists only of describing their platform and their aspirations. There is no scientific claim to review and nothing for a reviewer to do except to say that this manuscript doesn’t belong in a scientific journal.”

These reviewer comments suggest that the reviewer is not aware of the fact that PLOS ONE explicitly accepts Methods papers, and is unfamiliar with the guidelines by which such papers should be assessed. §2 describes the background to the field and existing alternatives. §3 describes our method (not our platform). 

“I will say that judging “classification accuracy” by treating “predictions >50% as predictions of replication success and <50% as predictions of replication failure” is extremely maladroit.”

As described in §2 and throughout, there are multiple measures for prediction accuracy. Classification accuracy is a standard metric within this field, is reported by previous authors, and our use of it allows us to benchmark our method's validity against the existing literature. We would happily describe in more detail the advantages and disadvantages of the various standard metrics for prediction outcomes.

“Why go to the trouble of eliciting probabilistic predictions?”

We have described in §3 and throughout the intrinsic advantages of probabilistic estimation, including reducing overconfidence and harnessing diverse viewpoints through mathematical rather than behavioural aggregation. We would happily go into more detail if these arguments were unclear.

---

## [Decision Letter · Decision Letter 1]

9 Feb 2022

PONE-D-21-05317R1Predicting reliability through structured expert elicitation with repliCATS (Collaborative Assessments for Trustworthy SciencePLOS ONE

Dear Dr. Bush,

Thank you for submitting your manuscript to PLOS ONE. After careful consideration, we feel that it has merit but does not fully meet PLOS ONE’s publication criteria as it currently stands. Therefore, we invite you to submit a revised version of the manuscript that addresses the points raised during the review process.

We look forward to receiving your revised manuscript.

Kind regards,

Ferrán Catalá-López

Academic Editor

PLOS ONE

Journal Requirements:

1. PLOS ONE has specific requirements for studies that are presenting a new
 method or tool as the primary focus (https://journals.plos.org/plosone/s/submission-guidelines#loc-methods-software-databases-and-tools.) One requirement is that the tool must meet the criteria of validation, which may be met by including a proof-of-principle experiment or analysis. To this effect, please ensure that you have described your pilot study in sufficient detail so as to adequately demonstrate that the new tool achieves its intended purpose.

"This project is sponsored by the Defense Advanced Research Projects Agency (DARPA) under cooperative agreement No.HR001118S0047. The content of the information does not necessarily reflect the position or the policy of the Government, and no official endorsement should be inferred."

4. Please upload a new copy of Figure 2 as the detail is not clear. Please follow the link for more information: " ext-link-type="uri" xlink:type="simple">https://blogs.plos.org/plos/2019/06/looking-good-tips-for-creating-your-plos-figures-graphics/"
" ext-link-type="uri" xlink:type="simple">https://blogs.plos.org/plos/2019/06/looking-good-tips-for-creating-your-plos-figures-graphics/"

5. We note that Figure 2 in your submission contain copyrighted images. All PLOS content is published under the Creative Commons Attribution License (CC BY 4.0), which means that the manuscript, images, and Supporting Information files will be freely available online, and any third party is permitted to access, download, copy, distribute, and use these materials in any way, even commercially, with proper attribution. For more information, see our copyright guidelines: http://journals.plos.org/plosone/s/licenses-and-copyright.

6. Please provide additional details regarding participant consent. In the Methods section, please ensure that you have specified (1) whether consent was informed and (2) what type you obtained (for instance, written or verbal). If your study included minors, state whether you obtained consent from parents or guardians. If the need for consent was waived by the ethics committee, please include this information.

Additional Editor Comments (if provided):

For further considering your manuscript at PLOS ONE, please see and address the comments and requests by two independent reviewers below.

PLOS ONE will consider submissions that present methods, as the primary focus of the manuscript if they meet specific criteria. Please, visit the journal website:

https://journals.plos.org/plosone/s/submission-guidelines#loc-methods-software-databases-and-tools

Reviewers' comments:

Reviewer's Responses to Questions

**Comments to the Author**

1. If the authors have adequately addressed your comments raised in a previous round of review and you feel that this manuscript is now acceptable for publication, you may indicate that here to bypass the “Comments to the Author” section, enter your conflict of interest statement in the “Confidential to Editor” section, and submit your "Accept" recommendation.

Reviewer #2: (No Response)

Reviewer #3: All comments have been addressed

2. Is the manuscript technically sound, and do the data support the conclusions?

Reviewer #2: No

Reviewer #3: Yes

3. Has the statistical analysis been performed appropriately and rigorously? 

Reviewer #2: No

Reviewer #3: N/A

4. Have the authors made all data underlying the findings in their manuscript fully available?

Reviewer #2: No

Reviewer #3: Yes

5. Is the manuscript presented in an intelligible fashion and written in standard English?

Reviewer #2: Yes

Reviewer #3: Yes

6. Review Comments to the Author

Reviewer #2: Dear Authors,

Thank you for the opportunity to review this manuscript. I realize there has been a round 1 and disagreements with one of the reviewers, and due to the deadline, that was given me – Feb 28, I am not sure of the effect of this review – as the form says the data will already be collected and deposited by March 2021 (possibly a typo) and in order to evaluate the correct accuracy and validity of the data, as the other reviewer said – I would need access to data that support these claims.

Nevertheless, here are some of my suggestions:

1) It is not clear when reading the abstract that this is a methods paper, and that it demonstrates utility, validation or addresses availability of the method or the platform. So please restructure the paper so that it has clear sections with methods and results for utility and validity, and comment on availability of the questions as well as the platform that you used.

2) If it is a method paper then full data, which demonstrate validity, feasibility, accuracy, and availability need to be made available for reviewers to evaluate them. If on the other hand this is a more a protocol or proof of concept – then details on how validity and utility will be assessed when it is finished need to be described in detail and pilot results can be moved to appendix and used only to say that with them that you showcase feasibility of the proposed assessment of utility / validity /accuracy.

2) Intro - replicable, in that repetitions of the methods and procedures should 4

produce the same facts/result – I believe its is more important to state here they are first reproducible – difference between reproducibility and replicability needs to be stressed in this paper. Additionally, as this project in a way evaluates also reproducibility of the papers – a clear description needs to be made on how reproducibility influences replicability.

3) Intro - I would move all text starting from -The elicitation procedure in this paper is based on the IDEA protocol – in the methods – explaining how the process will be done. End the intro with your objective for this paper.

4) While your first question was whether or not the claim will replicate – it sounds very much that this question translates to whether they believe the results of the study are reliable/valid. If there was time to amend your protocols, I would insist you ask this in one group of participants, before the question on replicability to see how often would answer to these two questions differ. Additionally, when a reviewer finds that the study would replicate – but it is nevertheless useless in its findings –the question remains of should resources be spent on its replication.

2) Research context – structure suggestion - the objectives stated in this section should be numbered and methods to assess each explained in detail.

3) I suggest moving the platform section in the appendix and focusing on the method for prediction exclusively, or make it clear if this is a paper about the platform – or about both.

4) As you stated that Full details of current performance are presented in Wintle et al. (in review) and Hanea et al.(in review) – it is not clear what was the purpose of those papers, vs this one. Which one represents validity / feasability / methods paper, and why is there a need for this paper if the methods are validated in other papers?

5) A validation method in my view requires use of one or more different methods on the same studies, and showcasing accuracy of your vs the other methods. A clear table with this is needed. If it about the platform, then user experience surveys, accessibility and other items need to be addressed. Please clearly state - what are you demonstrating with subtitles, and methods that you use for that.

6) While I appreciate the challenges section, please focus this paper on its primary goal – is it a protocol of your method, a methods paper, or a protocol on how full validation will be done. The purpose of reviewers should be to be able to say if your validation methods are sufficient to say this method is good for what you propose. The challenges could be renamed to advice to those planning to implement them, and you should consider creating a checklist, or flowchart that researchers could follow, but these sections should come only after we agree that you provided evidence of the proof of utility/validity/ superiority of your method . Similarly for future application. These sections are currently too long, and take away from the focus of the paper – you can move them to the appendix. Focus on the goal – to demonstrate that this method can be used to evaluate replicability. Include examples of discussions, reflections on time to provide feedback and similar aspects.

7) If in conclusions you say that you still do not have answer to whether repliCATS reliably improves replication prognostics – then you need to be clear what exactly do you mean you are validating in this paper under section 4. PLOS guidelines say demonstrate that the method achieves its intended purpose, and therefore structure the paper in that way. To me this read as methods paper with a proof of principle – and the tile and the structure should reflect that. Papers on its accuracy compared to other methods are pending, and that needs to be made clear in the abstract and in the main message of the paper. You could also consider structuring this paper in a way that would help others using your method to evaluate this particulars study’s replicability. Finally, please rewrite your conclusions and say – In this paper we showcase a new method or a new platform that can be used to …. We have also demonstrated its x and y. Our pilot data indicate z, with studies on a, b, c, pending. We also believe this method could be used for…

8) My final recommendation would be for the editor to provide you with an example of a methods paper published in PLOS or elsewhere according to which you would structure this paper. Alternatively, you could cite, in the begging a methods, template you followed. Currently the structure is not easy to follow nor clearly states the advantages and limitations of methods you used to demonstrate the purpose, validity, superiority.

Reviewer #3:

Authors have appealed against the initial decision to reject their paper. They argue that :

- There was no major issue raised by the reviewer ;

- That this is a method paper and therefore that the fact that there is no specific data/interpretation was appropriate ;

I must say that I agree with the authors. This is a paper describing a method. The main difficulty is that it reviews and summarize previous material / preliminary evidence (including preprints) suggesting that that this new method is appropriate and validate. The boundary between a method paper and a review is therefore small and I can also understand the opinion of the first editor and reviewer.

However, if one one follow PLOS One guidance, this manuscript reports data on :

- Utility ;

- Validity (although this is preliminary evidence) ;

- And, to a lesser extent on availabilty (i.e. material from various projects may available through the Open Science Framework) ;

Regarding this last point, the following data sharing statement is not acceptable regarding PLOS One policy :

"This paper describes a research methodology prior to final results. Not all results from the research have been finalised and so results cannot yet be made available.

The data underlying this research will be made available via an OSF repository after March 2021. Data will be anonymised before being made publicly available. Some participant data were collected on a confidential basis and cannot be made publicly available due to ethical concerns."

I mean, that as a reviewer of this specific paper, I don't care about data that is being collected using the method proposed by the authors of this paper (because this data will be necessary to share for the future papers reporting on the results) but I am interested in all material described in this paper. For instance, concerning this very relevant link that is cited in the paper (https://osf.io/m6gdp/). This is not publicly available and one need to ask for access. It needs to be made publicly available. No restrictions are acceptable.

Authors should provide, a detailed table reporting for each section of the manuscript :

- All links to registered protocols ;

- A link to aggregated results (e.g. publication, pre-prints) ;

- All links to datasets and code (when available) ;

Of course, in some occasions, this can be found in the text but a clear table detailling all the output and describing where data can be found would be great.

This was my major point regarding this paper. I have a few minor points :

- In the title make it explicit that it is a "METHOD" paper. You may want to add that you are elaborating on "UTILITY, VALIDITY and AVAILABILITY" ;

- Please avoid acronyms in the abstract :

"We introduce a new technique to evaluating replicability, the repliCATS (Collaborative Assessments for Trustworthy Science) process, a structured expert elicitation approach based on the IDEA protocol." A reader may have no idea of what the IDEA protocol is...

- In the abstract, again, please elaborate more on the 3 categories "UTILITY, VALIDITY and AVAILABILITY" and detail how each was investigated or addressed.

- Please also add in the abstract the main limitation of the approach to be fair as there are some limitations.

- Please elaborate a little bit more on the feasibility of claims like "the IDEA protocol for many experts addressing many problems, with 141 the capacity for the assessment of 3000 claims in 18 months."

- I do think that results of the pilot study may nicely be illustrated with a figure ;

- Regarding your answer to the reviewer : "We would happily describe in more detail the advantages and disadvantages of the various standard metrics for prediction outcomes." I agree and support you in working on a figure and or a table to detail this point.

- Last panels in Figure 2 are not appropriately ordered and the order should be a, b, c... I understand the restrictions in terms of space, but I really invite you to organise it better.

7. PLOS authors have the option to publish the peer review history of their article (what does this mean?). If published, this will include your full peer review and any attached files.

Reviewer #2: **Yes: **

Reviewer #3: **Yes: **

---

## [Author Response · Author response to Decision Letter 1]

20 Jun 2022

Copyright statement:

Neither Figure 1 nor 3 have been previously copyrighted nor contain any elements which are under copyright restriction. All elements of these figures were either generated by the authors of the manuscript, are images copyright for which resides with the authors of the manuscript, or are images that have been used under the licence CC0 1.0 Universal (CC0 1.0) Public Domain Dedication (‘No copyright’).

Responses to editor

“One requirement is that the tool must meet the criteria of validation, which may be met by including a proof-of-principle experiment or analysis.”

We have described in §2.5 more detail regarding the validation experiment and provided a new figure including results. While we regard the results of this as more than a proof-of-concept experiment, we have adjusted the wording of the paper to address this concern, shared by Reviewer 2. The SCORE program has been a large, complex, multi-disciplinary and multi-team project and it has not been possible to cover all of the project's aims, methods and results in a single paper. Moreover, some of the data from the overall program is still embargoed for public release, as the work of some teams is ongoing. However, all data underlying our validation experiment is now publicly available and we have included a link. 

“Please include this amended Role of Funder statement in your cover letter; we will change the online submission form on your behalf.”

Updated funder statement was provided in the previous cover letter. We will include it in this cover letter. 

“Should your manuscript be accepted for publication, we will hold it until you provide the relevant accession numbers or DOIs necessary to access your data.”

Data underlying the validation experiment reported in this paper is now publicly available and we have clarified this in re-submission. 

“Please upload a new copy of Figure 2 as the detail is not clear.”

New Figure 2 was uploaded with previous resubmission, and this new Figure was specifically referred to by Reviewer 3. Could the editor please confirm if further revision is required. 

“We note that Figure 2 in your submission contain copyrighted images”

All graphics in Figure 2 were created by the repliCATS project. Could the editor please confirm which graphics are of concern.

“Please provide additional details regarding participant consent.”

Accepted; additional details provided in §3 as requested. 

 

Responses to Reviewer 2

“So please restructure the paper so that it has clear sections with methods and results for utility and validity, and comment on availability of the questions as well as the platform that you used.”

Accepted; we have restructured as requested so that results on validity are made clear in §2.5. The precise elicitation we used is provided in the Supporting Information section. The process itself does not rely on our specific implementation; we have clarified in §3 that the description of our platform is provided as an example. 

“If it is a method paper then full data, which demonstrate validity, feasibility, accuracy, and availability need to be made available for reviewers to evaluate them.”

Accepted; we have included in §2.5 more details about the accuracy of our validation experiment and a link to our full data from this experiment.

“I believe its is more important to state here they are first reproducible”

Accepted; the term used in §1 Introduction, “reliable”, was intended to include computational reproducibility and we have clarified this. A full discussion of issues around computational reproducibility, however, is beyond the scope of this paper. 

“make it clear if this is a paper about the platform – or about both.”

Accepted; we have clarified in §3 that the online platform is just an example of a practical implementation of the method, which is based on the elicitation procedure plus aggregation methods. We have described elsewhere the technical implementation of the platform so that others could use that, but the method itself could be operationalised in many different ways to suit different contexts. 

“A validation method in my view requires use of one or more different methods on the same studies, and showcasing accuracy of your vs the other methods.”

Partly accepted; the studies evaluated by participants in the repliCATS project were also the subject of prediction experiments using other techniques. We use the reported results of those experiments for comparison. We have clarified this. 

“While I appreciate the challenges section, … . Similarly for future application. These sections are currently too long,”

Accepted; we have reduced the length of these sections. 

 

Responses to Reviewer 3

“this very relevant link that is cited in the paper (https://osf.io/m6gdp/). This is not publicly available and one need to ask for access. It needs to be made publicly available. No restrictions are acceptable.”

Accepted, and with our apologies. We included the private rather than the public link and this error has been fixed. We have confirmed the data availability. 

“A reader may have no idea of what the IDEA protocol is”

Accepted; change made as requested. We outline the basic characteristics of the protocol and direct the reader to previously published work that comprehensively describe it.

“In the abstract, again, please elaborate more on the 3 categories "UTILITY, VALIDITY and AVAILABILITY" and detail how each was investigated or addressed.”

Accepted; we emphasise that the “utility of processes to predict replicability is their capacity to test scientific claims without the costs of full replication” and in §3 that this method is available for a range of implementations, not only our implementation. 

“Please also add in the abstract the main limitation of the approach to be fair as there are some limitations.”

Accepted; major limitation added to abstract as requested.

“Please elaborate a little bit more on the feasibility of claims like "the IDEA protocol for many experts addressing many problems, with the capacity for the assessment of 3000 claims in 18 months.””

Accepted; more detail provided in §3 about the specific implementation within the SCORE project. 

“I do think that results of the pilot study may nicely be illustrated with a figure”

Accepted; figure provided in §2.5 as requested.

“Regarding your answer to the reviewer: "We would happily describe in more detail the advantages and disadvantages of the various standard metrics for prediction outcomes." I agree and support you in working on a figure and or a table to detail this point.”

Accepted; table added in in §2.2 as requested. 

“Last panels in Figure 2 are not appropriately ordered and the order should be a, b, c... I understand the restrictions in terms of space, but I really invite you to organise it better.”

Not accepted; table is labelled anticlockwise from centre top. We believe this is a straightforward and readily comprehensible way to organise the information. This has been clarified in the figure caption.

---

## [Decision Letter · Decision Letter 2]

6 Jul 2022

PONE-D-21-05317R2Predicting reliability through structured expert elicitation with the repliCATS (Collaborative Assessments for Trustworthy Science) processPLOS ONE

Dear Dr. Bush,

Thank you for submitting your manuscript to PLOS ONE. After careful consideration, we feel that it has merit but does not fully meet PLOS ONE’s publication criteria as it currently stands. Therefore, we invite you to submit a revised version of the manuscript that addresses the points raised during the review process.

If applicable, we recommend that you deposit your laboratory protocols in protocols.io to enhance the reproducibility of your results. Protocols.io assigns your protocol its own identifier (DOI) so that it can be cited independently in the future. For instructions see: https://journals.plos.org/plosone/s/submission-guidelines#loc-laboratory-protocols. Additionally, PLOS ONE offers an option for publishing peer-reviewed Lab Protocol articles, which describe protocols hosted on protocols.io. Read more information on sharing protocols at https://plos.org/protocols?utm_medium=editorial-emailutm_source=authorlettersutm_campaign=protocols.

We look forward to receiving your revised manuscript.

Kind regards,

Ferrán Catalá-López

Academic Editor

PLOS ONE

Additional Editor Comments (if provided):

Dear Dr. Bush,

Thank you very much for your improved manuscript. Before I can consent to publication in PLOS ONE, however, I still have the following issues you need to deal with.

Points:

• Methodological reports must meet the criteria of utility, validation, and availability, which are described in detail at https://journals.plos.org/plosone/s/submission-guidelines

• Specifically, authors should adequately describe the validation of their methods in the pilot study, and whether they have included sufficient detail on the results of the study.

• Authors’ should provide detailed responses to the peer reviewers’ comments (specially, reviewer 2) and consider the additional requests for your improved manuscript.

My decision to recommend publication will depend on how we get these issues solved and worked into your fine manuscript. By personal experience, I know that this is not the answer you wanted. But before you feel dishearten I can assure you that these issues should substantially improve the messages your manuscript send to the potential readers as well as make the text more coherent.

Thank you.

Reviewers' comments:

Reviewer's Responses to Questions

**Comments to the Author**

1. If the authors have adequately addressed your comments raised in a previous round of review and you feel that this manuscript is now acceptable for publication, you may indicate that here to bypass the “Comments to the Author” section, enter your conflict of interest statement in the “Confidential to Editor” section, and submit your "Accept" recommendation.

Reviewer #2: (No Response)

Reviewer #3: All comments have been addressed

2. Is the manuscript technically sound, and do the data support the conclusions?

Reviewer #2: No

Reviewer #3: Yes

3. Has the statistical analysis been performed appropriately and rigorously? 

Reviewer #2: No

Reviewer #3: N/A

4. Have the authors made all data underlying the findings in their manuscript fully available?

Reviewer #2: No

Reviewer #3: Yes

5. Is the manuscript presented in an intelligible fashion and written in standard English?

Reviewer #2: Yes

Reviewer #3: Yes

6. Review Comments to the Author

Reviewer #2: Thank you for inviting me to look at this again. I dislike that the authors did not provide full copy of my previous comments, nor that I received a message from the editor on what is the expected and agreed upon structure for this paper. The following are changes I would like to see before the paper is published:

1) Please rewrite the abstract, to showcase actual data/numbers on Utility, Accuracy, Scalability, Insight and Validity

2) Please expand your definition of reproducibility in the intro. It is not only about the fact that “ multiple observers examining the same data should agree on the facts and the results of analyses” – it is also about the reproducibility of the manuscript itself, and all data/figures produced in it – this aspect requires more attention in the introduction and in the discussion – and its relation to replicability. Additionally, authors should make clear examples of disciplines that replicate their own findings in the primary articles – and therefore do not require replication to be done after study publication. If the culture change would require replications to be included in the publication of a primary study - then the evaluation of strength of claims would be dfferent then they are explained here. Please mention examples of papers and fields that already do this, and why this is not necessary or feasabile for all types of studies. Reproducible manuscript are however feasible.

3) Please be consistent – you stated in intro: describe the utility, accuracy and scalability of this method – and yet the section 2 is titled - Utility, Accuracy, Scalability, Insight and Validity. If all will be described, that should be stated.

4) I am puzzled again by section 2 - utility should be about the utility of repliCATS project not replication studies themselves. Same applies to the rest. I would advise arranging a call with the editor and agreeing on the structure of this paper.

5) Section 2.2 did not specify what is the accuracy of repliCATS vs prediction markets, vs surveys – therefore this section also requires rewriting. If as I mentioned in the round 1 review, this paper is about how accuracy will be measured for repliCATS – then this paper needs to be turned into a protocol for all items in section 2.

6) I strongly suggest authors create a table where in the rows are Utility, Accuracy, Scalability, Insight and Validity and in 3 columns – surveys, predication markets, and repliCATS project – additional columns can be added – to include actual replication studies or other methods authors believe should be included. And then the paper should be structured in such a way that all of these are covered. Until this is done, I will refrain for making comments on the rest of the manuscript.

Reviewer #3: Thank you for addressing my comments. I think the manuscript has improved now and is suitable for publication.

7. PLOS authors have the option to publish the peer review history of their article (what does this mean?). If published, this will include your full peer review and any attached files.

Reviewer #2: **Yes: **Mario Malicki

Reviewer #3: **Yes: **Florian NAUDET

---

## [Author Response · Author response to Decision Letter 2]

25 Aug 2022

This text is also included in the response to reviewers document. 

In this revision, only Reviewer 2 has requested changes. We accept Reviewer 2’s points 1, 3, 5 and partially point 4. We do not accept Reviewer 2’s points 2 and 6. Details are provided below. 

Responses to Reviewer 2

1) Please rewrite the abstract, to showcase actual data/numbers on Utility, Accuracy, Scalability, Insight and Validity

We accept this recommendation. The abstract has been re-written to include specific figures for Accuracy and Validity, and Scalability. We note that Utility and Insight are qualitative features of the repliCATS process rather than quantitative ones and so specific figures are not included. 

2) Please expand your definition of reproducibility in the intro. It is not only about the fact that “ multiple observers examining the same data should agree on the facts and the results of analyses” – it is also about the reproducibility of the manuscript itself, and all data/figures produced in it – this aspect requires more attention in the introduction and in the discussion – and its relation to replicability. Additionally, authors should make clear examples of disciplines that replicate their own findings in the primary articles – and therefore do not require replication to be done after study publication. If the culture change would require replications to be included in the publication of a primary study - then the evaluation of strength of claims would be dfferent then they are explained here. Please mention examples of papers and fields that already do this, and why this is not necessary or feasabile for all types of studies. Reproducible manuscript are however feasible.

We do not accept this recommendation. We do not agree with Reviewer 2 that reproducibility is necessarily a prior goal in the assessment of scientific manuscripts, nor is the discussion about practices in different fields in scope for this paper with a well-defined disciplinary range. This recommendation appears to be based on a specific opinion of Reviewer 2, which we do not share, and we do not feel we need to rewrite our manuscript accordingly. We have, however, substantially rewritten the second paragraph of the paper (in lines 9 – 13) to clarify the scope even more explicitly. 

3) Please be consistent – you stated in intro: describe the utility, accuracy and scalability of this method – and yet the section 2 is titled - Utility, Accuracy, Scalability, Insight and Validity. If all will be described, that should be stated.

We accept the recommendation and thank the reviewer for noting the inconsistency in the heading to Section 2. This has been fixed (lines 52-53).

4) I am puzzled again by section 2 - utility should be about the utility of repliCATS project not replication studies themselves. Same applies to the rest. I would advise arranging a call with the editor and agreeing on the structure of this paper.

We partially accept this recommendation. While we believe we had discussed the cost-saving utility of the repliCATS (and similar) processes, we have added additional detail to Section 2.1 in lines (64-66) to make this even more explicit. 

5) Section 2.2 did not specify what is the accuracy of repliCATS vs prediction markets, vs surveys – therefore this section also requires rewriting. If as I mentioned in the round 1 review, this paper is about how accuracy will be measured for repliCATS – then this paper needs to be turned into a protocol for all items in section 2.

As discussed, we have restructured Section 2 (lines 67-155) to combine Accuracy and Utility into a single sub-section. 

6) I strongly suggest authors create a table where in the rows are Utility, Accuracy, Scalability, Insight and Validity and in 3 columns – surveys, predication markets, and repliCATS project – additional columns can be added – to include actual replication studies or other methods authors believe should be included. And then the paper should be structured in such a way that all of these are covered. Until this is done, I will refrain for making comments on the rest of the manuscript.

We do not accept this recommendation. This would require a complete re-write of the manuscript according to the stylistic preference of Reviewer 2 and we do not consider this to be a reasonable request.

---

## [Editor Report · Decision Letter 3]

30 Aug 2022

Predicting reliability through structured expert elicitation with the repliCATS (Collaborative Assessments for Trustworthy Science) process

PONE-D-21-05317R3

Dear Dr. Bush,

We’re pleased to inform you that your manuscript has been judged scientifically suitable for publication and will be formally accepted for publication once it meets all outstanding technical requirements.

Kind regards,

Ferrán Catalá-López

Academic Editor

PLOS ONE

Additional Editor Comments (optional):

Thank you for this revised version of your manuscript, and for your detailed answers to reviewer(s) suggestions.
---

## [Editor Report · Acceptance letter]

19 Sep 2022

PONE-D-21-05317R3 

Predicting reliability through structured expert elicitation with the repliCATS (Collaborative Assessments for Trustworthy Science) process 

Dear Dr. Bush:

I'm pleased to inform you that your manuscript has been deemed suitable for publication in PLOS ONE. Congratulations! Your manuscript is now with our production department. 

Kind regards, 

on behalf of

Dr. Ferrán Catalá-López 

Academic Editor

PLOS ONE